# Multi-modal Transfer Learning between Biological Foundation Models

**Juan Jose Garau-Luis**[*]
InstaDeep

**Patrick Bordes**[*]
InstaDeep

**Liam Gonzalez**[*]
InstaDeep

**Masa Roller**
InstaDeep

**Bernardo P. de Almeida**
InstaDeep

**Lorenz Hexemer**
BioNTech

**Christopher Blum**
BioNTech

**Stefan Laurent**
BioNTech

**Jan Grzegorzewski**
BioNTech

**Maren Lang**
BioNTech

**Thomas Pierrot**[†]
InstaDeep

**Guillaume Richard**[†]
InstaDeep

## Abstract

Biological sequences encode fundamental instructions for the building blocks of life, in the form of DNA, RNA, and proteins. Modeling these sequences is key to understand disease mechanisms and is an active research area in computational biology. Recently, Large Language Models have shown great promise in solving certain biological tasks but current approaches are limited to a single sequence modality (DNA, RNA, or protein). Key problems in genomics intrinsically involve multiple modalities, but it remains unclear how to adapt general-purpose sequence models to those cases. In this work we propose a multi-modal model that connects DNA, RNA, and proteins by leveraging information from different pre-trained modality-specific encoders. We demonstrate its capabilities by applying it to the largely unsolved problem of predicting how multiple RNA transcript isoforms originate from the same gene (i.e. same DNA sequence) and map to different transcription expression levels across various human tissues. We show that our model, dubbed IsoFormer, is able to accurately predict differential transcript expression, outperforming existing methods and leveraging the use of multiple modalities. Our framework also achieves efficient transfer knowledge from the encoders pre-training as well as in between modalities. We open-source our model, paving the way for new multi-modal gene expression approaches.

## 1 Introduction

Foundation models have ignited a revolution in numerous scientific fields, starting in NLP and computer vision, and more recently in several domains within the life sciences. Within the biological sciences, these models have enabled predicting protein structures from sequences [1, 2], deciphering the genome functions [3, 4, 5, 6] and interactions of biomolecules [7], and crafting new molecules not found in nature [8]. Specifically, significant progress has been made with foundation models tailored to DNA [3, 4, 5, 9], RNA [10, 11, 12, 13], and protein [2] sequences. These foundation models have typically been developed and trained separately, using either self-supervised learning techniques

---

[*]Equal Contribution
[†]Equal Supervision

38th Conference on Neural Information Processing Systems (NeurIPS 2024).

such as Masked Language Modeling (MLM), as seen in models like ESM [2] and the Nucleotide Transformer (NT) [3], or supervised learning approaches on large datasets, as in AlphaFold [1] and Enformer [14]. These models have been instrumental in advancing our understanding of biology by accurately predicting the structures and functions of biological sequences.

While existing methods provide relevant insights, they are still limited by the fact that they only consider a single sequence modality. In biology, the *central dogma* describes the flow of genetic information from DNA to RNA to proteins [15]. This fundamental concept underscores the interconnectedness of these three types of biological sequences and highlights the potential for a unified modeling approach. An architecture that integrates DNA, RNA, and protein modalities should provide a comprehensive model for biology that mirrors the natural processes within cells. Furthermore, by enabling transfer learning across modalities, the model can capitalize on the vast amounts of pre-training already performed on individual DNA, RNA, and protein datasets.

Developing deep learning models using multiple biological sequence modalities has been mainly limited by the lack of matched available data; existing databases usually isolate a specific modality and thus relationships between modalities are not easily obtainable. As more multi-modal datasets are made available [16, 17], it is becoming possible to develop models that extract and combine the information from DNA, RNA, and protein sequences to better model the different cellular processes. Such multi-modal models have already been successful in other domains, such as mixing language and visual inputs [18, 19, 20, 21, 22, 23, 24, 25], but until now there are no models that can handle multiple biological sequence modalities.

In our work, we propose the first multi-modal architecture to connect DNA, RNA, and proteins (Fig. 1). Our approach is based on three main components: (i) pre-trained modality-specific encoders that produce one embedding per modality, (ii) aggregation layers that combine information from the encoders and create a multi-modal representation, and (iii) a task-specific head that predicts the desired output. We show that our multi-modal approach transfers and aggregates knowledge of pre-trained mono-modal encoders and outperforms previous single-modality baselines (Enformer [14], NT [3], and ESM [2]). We also demonstrate the flexibility of our approach by comparing different encoders for a specific modality and different aggregation techniques. While some previous approaches have modeled specific interactions between modalities, such as protein-to-DNA interaction using structure information and modules [26], our approach is general-purpose and can be adapted to any task involving one or more biological sequence types.

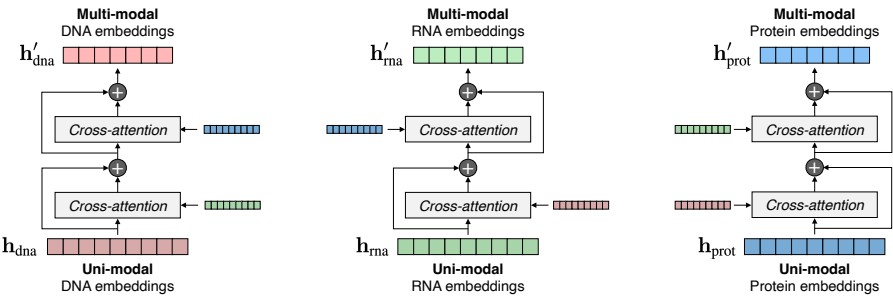

Figure 1: Our aggregation module compiles information from the different biological sequence modalities of DNA, RNA, and proteins by using successive cross-attention layers and residual connections.

Significant problems in genomics intrinsically involve multiple sequence modalities [27, 26], and it is still unclear how to adapt general-purpose sequence models to those cases. In order to validate our multi-modal approach, we focused on the study of a crucial task in genomics that has been challenging to tackle using a single sequence modality - namely the prediction of RNA *transcript isoform expression* across different tissues [28]. When gene DNA sequences are transcribed into mRNA molecules to produce proteins, they generally do not express it in just a single way producing a single protein isoform. After DNA sequences are transcribed, pre-mRNA transcripts undergo a process called RNA alternative splicing where they are cut and re-assembled to form different variants of mature mRNA molecules that can be translated into proteins. This process allows for the generation of multiple RNA and protein isoforms from a single gene DNA sequence, that can differ in structure, function, localization, or other properties (Fig. 2a). Therefore, predicting which isoforms are expressed in a given cell or under specific conditions is key to understand gene regulation and disease mechanisms.

This task is multi-modal in nature, since only looking at the DNA sequence present in the cell does not provide a complete picture of the different RNA isoform landscape. Focusing on this single important task in opposition to tackling more but less challenging tasks should provide a stronger evidence for the effectiveness of our approach to advance biological knowledge.

In summary, our contributions are the following: (i) We built the first multi-modal model for the integration of DNA, RNA, and protein sequences. (ii) We show that our model achieves efficient transfer learning between the three modalities, not only leveraging intra-modalities pre-training but also inter-modalities transfer. (iii) We use our architecture to tackle a new central task in biology that requires a multi-modal approach, namely RNA *transcript isoform expression prediction*, and obtain state-of-the-art results, overcoming limitations of existing gene expression models such as Enformer. (iv) Finally, we performed ablation studies to validate our different architectural choices and release our trained isoform expression prediction model IsoFormer, providing a new framework and baseline to the community and opening the door to future research on multi-modal sequence modeling and multi-modal biological problems.

## 2 Related Work

**Biological sequence modeling**    Researchers have explored different ways to process DNA, RNA, and protein sequences for multiple applications. Initial approaches included dynamic programming [29], hidden Markov models [30], and genomic hash tables [31]. Recently, deep learning, via supervised [1] and unsupervised frameworks [3, 4, 13, 2], has gained thrust in the community. These methods, influenced by advancements in Large Language Models [32, 33], have focused on DNA [3, 4, 5, 9], RNA [10, 11, 12, 13], and protein sequences [2]. They capture complex biological patterns and perform tasks like protein structure prediction [2] and variant effect prediction [34]. These efforts have become more targeted given the different challenges of each sequence modality. For instance, models like HyenaDNA [5, 9] or Caduceus [35] tackled the long-range dependencies in DNA. Few models consider multiple modalities, mainly focusing on structural information (e.g. predicting DNA interacting residues in proteins [26]). Our work, IsoFormer, is the first general-purpose model integrating three biological sequence modalities.

**Multi-Modal Integration**    Efforts to address multi-modality in deep learning models have been prominent in NLP [36, 37, 38, 39, 40] and computer vision [41, 42, 43, 44, 45]. In computer vision, integrating image and audio for video classification [46] or audio-visual segmentation [47] is common, often using cross-attention [48]. Unsupervised approaches using contrastive learning also integrate image, audio, and other modalities [49]. Large Language Models have driven multi-modal efforts for text and image with methods like Flamingo [19] and CLIP [18], leading to various integration architectures, including the Perceiver Resampler [50, 19] and C-Abstractor [51, 52]. IsoFormer integrates DNA, RNA, and protein sequences, leveraging past integration architectures.

**Gene Expression Prediction**    Transcript isoform expression prediction is a more refined and challenging task within gene expression prediction. Traditionally, gene expression has been addressed through tailored approaches and experimental annotations [53, 54]. Recently, deep learning models have demonstrated improved results by predicting gene expression directly from DNA sequence [55, 56]. Enformer [14] leveraged dilated convolutions and attention layers to extend the context window to 190 kilo base pairs (kbp), achieving new state-of-the-art results on gene expression. However, these models are limited to gene-level expression levels and cannot predict isoform-specific expression as DNA sequence information is not sufficient to solve this task. We introduce IsoFormer, the first multi-modal model that is able to predict transcript isoform expression.

## 3 Background

**Central dogma of biology**    In this work, we consider three biological sequence types: DNA, RNA, and proteins. These sequences (composed of nucleotides or amino-acids) are fundamental to biological processes in living organisms. They are also intricately intertwined: DNA dictates RNA synthesis during *transcription*, and RNA guides protein synthesis through *translation* (see Fig. 2; [15]). Thus, changes in DNA can alter RNA and protein sequences, impacting an organism's phenotype (i.e. function). DNA and RNA sequences consist of four nucleotides (ACGT and ACGU, respectively), while proteins are made of 20 amino-acids. Thus, these sequences are commonly modelled as linear strings.

Obtaining data in this format is becoming more available thanks to recent advances in high-throughput sequencing technologies.

**Gene expression and isoforms**   Gene expression is the process by which a gene's DNA sequence is transcribed into various RNA molecules that code for specific proteins. This process is complex, as a single gene can produce different RNA and protein *isoforms* with varying abundances across tissues (Fig. 2a). RNA isoforms are mRNA molecules of different exon compositions derived from the same gene, produced via processes like alternative splicing. The expression level of each isoform is commonly measured by counting the amount of its RNA molecules in cells. Accurate isoform expression prediction across tissues can aid in understanding genetic variants' effects on cellular processes and phenotypes. However, this task cannot be tackled solely using DNA sequences as the same DNA sequence produces different isoforms in different cellular contexts and types. In addition, both DNA and RNA sequence features are crucial as DNA sequences contain regulatory elements that control transcription levels, while RNA isoform sequences have features affecting their stability and degradation.

## 4   Method

### 4.1   Multi-modal framework

We consider respectively DNA, RNA, and protein sequences $\mathbf{x}_{\text{dna}} \in \mathcal{A}_{\text{dna}}$, $\mathbf{x}_{\text{rna}} \in \mathcal{A}_{\text{rna}}$ and $\mathbf{x}_{\text{prot}} \in \mathcal{A}_{\text{prot}}$ where $\mathcal{A}_{\text{dna}}$ is the DNA base alphabet ACGT, $\mathcal{A}_{\text{rna}}$ is the RNA base alphabet ACGU, and $\mathcal{A}_{\text{prot}}$ is the set of 20 amino-acids. We then consider three associated modality-encoders $f$ with respective weights $(\theta_{\text{dna}}, \theta_{\text{rna}}, \theta_{\text{prot}})$ that encode sequences $\mathbf{x}$ into corresponding embeddings $\mathbf{h}$:

$$\mathbf{h}'_{\text{dna}} = f^{\text{agg}}_{\phi} \left( \mathbf{h}_{\text{dna}}, \mathbf{h}_{\text{rna}}, \mathbf{h}_{\text{prot}} \right), \quad \mathbf{h}'_{\text{rna}} = f^{\text{agg}}_{\phi} \left( \mathbf{h}_{\text{rna}}, \mathbf{h}_{\text{dna}}, \mathbf{h}_{\text{prot}} \right), \quad \mathbf{h}'_{\text{prot}} = f^{\text{agg}}_{\phi} \left( \mathbf{h}_{\text{prot}}, \mathbf{h}_{\text{dna}}, \mathbf{h}_{\text{rna}} \right) \quad (1)$$

We assume that the weights $(\theta_{\text{dna}}, \theta_{\text{rna}}, \theta_{\text{prot}})$ have been obtained through independent pre-training processes that can involve supervised or self-supervised training techniques over large corpus of biological data. Note that in practice, as commonly used in recent works [57], models pre-trained over DNA sequences can be re-used to produce embeddings for RNA sequences, replacing artificially the uracil base (U) by thymine (T) in the input. In this work, we aim to connect these encoders and train them jointly to learn a multi-modal embedding $\mathbf{h}_{\text{multi}}$. We start by learning a multi-modal embedding per modality defined as

$$\mathbf{h}'_{\text{dna}} = f^{\text{agg}}_{\phi} \left( \mathbf{h}_{\text{dna}}, \mathbf{h}_{\text{rna}}, \mathbf{h}_{\text{prot}} \right), \quad \mathbf{h}'_{\text{rna}} = f^{\text{agg}}_{\phi} \left( \mathbf{h}_{\text{rna}}, \mathbf{h}_{\text{dna}}, \mathbf{h}_{\text{prot}} \right), \quad \mathbf{h}'_{\text{prot}} = f^{\text{agg}}_{\phi} \left( \mathbf{h}_{\text{prot}}, \mathbf{h}_{\text{dna}}, \mathbf{h}_{\text{rna}} \right) \quad (2)$$

where $f^{\text{agg}}_{\phi}$ is an aggregation function with weights $\phi$. Then, we define the multi-modal embedding as the concatenation of the per modality multi-modal embeddings:

$$\mathbf{h}_{\text{multi}} = \left[ \mathbf{h}'_{\text{dna}}, \mathbf{h}'_{\text{rna}}, \mathbf{h}'_{\text{prot}} \right]. \quad (3)$$

Note that this definition is general as it allows the use of any aggregation function over the modality embeddings $\mathbf{h}$. The multi-modality embeddings per modality $\mathbf{h}'$ are introduced to solve tasks that are "modality-centered". For instance, $\mathbf{h}'_{\text{dna}}$ could be used to solve a task that involves nucleotide-level annotation over a DNA sequence while requiring other modalities as input. We rely otherwise on the concatenated multi-modal embedding $\mathbf{h}_{\text{multi}}$ to solve any other task.

### 4.2   Genes and isoforms expression

We now introduce our notations specific to the task of RNA isoform expression prediction (Fig. 2a). We consider a DNA sequence $\mathbf{x}_{\text{dna}}$ of length $L$ to contain a gene $g$. In practice, given the input size limitation of the existing foundation models, the sequence length $L$ might be shorter than the full length of the gene. In this case, we choose the DNA sequence $\mathbf{x}_{\text{dna}}$ to be centered on the start of the gene, i.e., where transcription begins, which also surrounds the promoter regions known to be important for transcription. This way, we also capture all the enhancer regulatory elements upstream of the transcription site.

We denote by $\mathbf{x}_{\text{rna}}^{(1)}, ..., \mathbf{x}_{\text{rna}}^{(n)}$ the $n$ existing transcripts for gene $g$ across all tissues. Coding transcripts are translated into proteins and we denote by $\mathbf{x}_{\text{prot}}^{(i)}$ the amino-acid sequence of the protein associated

to the transcript $\mathbf{x}_{\mathrm{rna}}^{(i)}$. We define the expression $e$ of genes and transcripts across tissues $T$ as:

$$\forall T, e\left(\mathbf{x}_{\mathrm{dna}}, T\right) = \sum_{i=1}^{n} e\left(\mathbf{x}_{\mathrm{rna}}^{(i)}, T\right) \in \mathbb{R}. \tag{4}$$

While deep learning models have been trained to predict the overall expression of genes across tissues with great accuracy [55, 56, 14, 58], to our knowledge no model can predict the expression of the different RNA transcripts across tissues directly from the sequence. In this work, we leverage our multi-modal framework to train a transcript expression level prediction model, dubbed IsoFormer, that takes as input a DNA sequence, an RNA transcript sequence, and its matching protein sequence to predict the expression of that transcript across tissues as measured by bulk RNA-seq (Fig. 2).

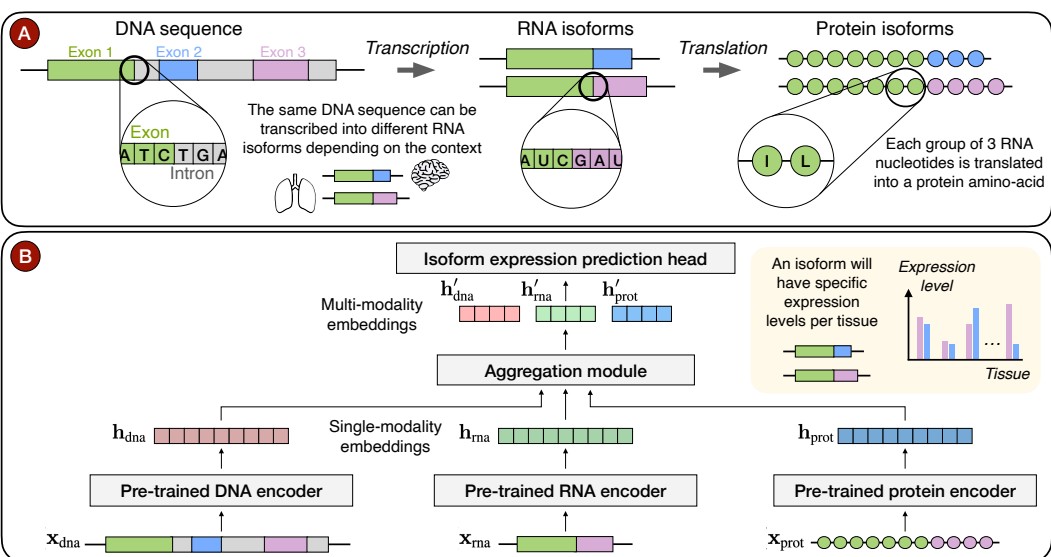

Figure 2: **a)** Three types of biological sequences are considered in this work: DNA, RNA, and proteins. These sequences are composed of nucleotides (DNA and RNA) or amino-acids (protein). In a single gene, several coding regions or exons can be used to create different RNA transcript isoforms and proteins. The abundance of each isoform is tissue-dependent and its measurement is called expression level. **b)** IsoFormer leverages pre-trained encoders that produce modality-specific embeddings, which are then aggregated into multi-modal embeddings. These are used to predict the expression of a given RNA transcript isoform across multiple tissues.

**Aggregation module** To capture the local patterns and their relationship across modalities, we introduce an embedding aggregation method based on cross-attention with residual connections (Fig. 1). The module is applied to each modality, performing cross-attention successively to the other modalities to produce multi-modal embeddings that are specific to each modality by keeping their dimensionality and which can be also stacked. Note that this aggregation method is robust to the absence of a specific modality as the cross-attention term will be zeroed out. The final multi-modal embedding $\mathbf{h}_{\mathrm{multi}}$ can then be used to solve any task. Our model can be trained end-to-end and it could be applied to different tasks by changing the prediction head.

## 4.3 Transferring from pre-trained biological encoders

Table 1: Characteristics of single-modality encoders considered in IsoFormer. † This column indicates the tokenization scheme (e.g., 1 token corresponds to 6 nucleotides).

| Modality | Model | Pre-training | Num. Params. | Tokens$^{\dagger}$ | Sequence Length |
|---|---|---|---|---|---|
| DNA | Enformer [14] | Supervised | 110M | 1 Nucleotide | 190,000 |
| DNA | NT (v2) [3] | Self-Supervised | 250M | 6 Nucleotides | 12,282 |
| RNA | NT (v2) [3] | Self-Supervised | 250M | 6 Nucleotides | 12,282 |
| Protein | ESM2 [2] | Self-Supervised | 150M | 1 Amino-acid | 2,047 |

**Encoders** We describe here the encoders used to process each sequence modality (see also Table 1 and Fig. 2b). For protein sequences, we used ESM (ESM2-150M) [2], which handles any protein up to 2,048 amino-acids. The ESM models have been pre-trained through MLM on large corpuses of protein sequences and are considered state-of-the-art on multiple tasks including folding. Protein sequences were all within ESM input length, so no additional processing was required. For DNA sequences, we used the NT model [3], which has also been trained through self-supervision to reconstruct masked 6-mers within 12 kbp genomic regions from 850 species. Additionally to the NT, we also considered the Enformer model as DNA encoder. The Enformer is a different type of model that has been pre-trained through full supervision to predict multiple experiments related to chromatin accessibility and modifications, transcription factor binding, and more importantly gene expression. Enformer is a strong candidate for our architecture's DNA encoder module as it can process sequences up to 200kbp as well as one can expect to obtain transfer from its gene expression capabilities. Finally, while foundation models have been reported to be pre-trained on RNA sequences, none of them has been made publicly available at the moment and pre-training a foundation model from scratch is out of the scope of this study. As such, we re-used the NT model [3] to compute embeddings for RNA sequences as this model has been reported recently to also be able to solve RNA [57] and protein [59] tasks with simple adaptations, using the trick described in section 4.1. RNA transcripts longer than 12kb were left-cropped to 12kb to conserve the 3'UTR regions, which are crucial for mRNA stability and polyadenylation, impacting isoform abundance. This adjustment affected only a small portion of the dataset.

**Architecture** Our architecture leverages three pre-trained encoders, one per bio-modality, as well as the aggregation function defined above, to generate a multi-modality embedding $\mathbf{h}_{\text{multi}}$ (Fig. 2b). That embedding is finally transformed by an expression head $f_\psi^{\text{exp}}$ with weights $\psi$ to make isoform expression level predictions across tissues. As the shape of the network output must be independent from the dimension of its inputs, we used a global average pooling over the length of the embedding. A linear layer then outputs one value per tissue. As long as the encoders can take in a biological sequence to produce an embedding, our method can function with different types of general-purpose encoders. While this flexibility allows us to leverage a big part of the landscape of biological sequence encoders, for this work we chose the specific models described in the section below.

## 4.4 Training

**Objective** We denote the IsoFormer model by $f_{\theta,\phi,\psi}$ where $\theta$ is the concatenation of the weights of the encoder models. These weights are initialized to the values obtained after pre-training of the different encoders. Respectively $\phi$ and $\psi$ denote the weights of the aggregation module and expression prediction head. The IsoFormer is trained to minimize the following objective

$$\mathcal{L}_{\text{MSE}} = \sum_T \left( f_\psi^{\text{exp}} \left( \mathbf{h}_{\text{multi}}, T \right) - e \left( \mathbf{x}_{\text{rna}}^{(i)}, T \right) \right)^2, \text{wherfe } \mathbf{h}_{\text{multi}} = f_{\theta,\phi}^{\text{enc}} \left( \mathbf{x}_{\text{dna}}, \mathbf{x}_{\text{rna}}^{(i)}, \mathbf{x}_{\text{prot}}^{(i)} \right) \quad (5)$$

where the summation is performed over a set of available tissues. Note that this framework can also accept only one or two of the three modalities as input. This is the case for instance when predicting expression of non-coding transcripts that do not translate into proteins.

**Dataset** We conducted our analysis of IsoFormer on RNA transcript expression data obtained from the GTEx[3] portal. We used Transcript TPMs measurements across 30 tissues, which come from more than 5,000 individuals. We followed a common process in gene expression datasets [56]: we averaged the expression levels for a given tissue across individuals, and used the reference genome sequence as input. We mapped transcripts to their original genes and associated proteins using the Ensembl database [60]. Our resulting dataset is made of triplets of RNA transcript sequences, DNA sequences (centered on the Transcription Start Site (TSS) of the transcript), and proteins. In total, the dataset is made of $\sim$170k unique transcripts, of which 90k are protein-coding and correspond to $\sim$20k unique genes. Our dataset has a fixed train and test set, divided by genes; all presented results correspond to the performance on the test set. We provide more details on the dataset in Appendix A.

**Hyperparameters** We used the Adam optimizer with a learning rate of $3 \cdot 10^{-5}$ and batch size of 64, and used early stopping on a validation set comprised of $5\%$ of the train set to reduce training time. We also made our baseline model's weights available[4]. More details in Appendix B.

---

[3] https://www.gtexportal.org/home/downloads/adult-gtex/bulk_tissue_expression
[4] https://huggingface.co/InstaDeepAI/isoformer

# 5 Experiments

We present extensive experiments to assess the performance of our multi-modal approach on the *transcript isoform expression prediction* task and compare it with existing single-modality approaches. We show that (i) our architecture efficiently aggregates modalities to improve its performance on this task; (ii) by using a tailored model for expression prediction as a base DNA encoder, IsoFormer reaches state-of-the-art performance; (iii) we provide an extensive ablation study on different aggregation approaches; (iv) we demonstrate that our approach achieves transfer learning both intra-modalities from their independent pre-training as well as inter-modalities.

## 5.1 Bridging three foundational models outperforms mono-modal approaches

**Experiment** We investigated the effect of adding the different modalities within our multi-modal framework for the prediction of expression of each RNA transcript across different human tissues. We used NT as the foundation model for both DNA and RNA and ESM as the protein encoder. We compared our multi-modal approach (DNA + RNA + protein) with models trained with different combinations of modalities as input: DNA only, RNA only, protein only, DNA + protein and DNA + RNA. Results were obtained over 5 random seeds; for each random seed we change the validation set and randomly initialize the non pre-trained parameters $(\phi, \psi)$ of our model. We report both $R^2$, which measures how well each model predicts the actual values of expression, and Spearman correlation across tissues, which is a metric for ranking transcripts based on their expression in each tissue.

Table 2: Performance of different variants of IsoFormer for the prediction of transcript isoform expression. $R^2$ and Spearman correlation across tissues for 5 different random seeds is reported. NT is used as both DNA and RNA encoder while ESM is used to process protein sequences.

| Model Input | $R^2$ | Spearman |
|---|---|---|
| DNA only | $0.13 \pm 0.02$ | $0.43 \pm 0.01$ |
| RNA only | $0.36 \pm 0.03$ | $0.61 \pm 0.01$ |
| Protein only | $0.20 \pm 0.01$ | $0.46 \pm 0.01$ |
| DNA + Protein | $0.28 \pm 0.01$ | $0.52 \pm 0.01$ |
| DNA + RNA | $0.39 \pm 0.01$ | $0.64 \pm 0.01$ |
| DNA + RNA + Protein | $\mathbf{0.43 \pm 0.01}$ | $\mathbf{0.65 \pm 0.01}$ |

**Results** We observed that our approach benefits from adding more modalities as the performance increases from one modality alone to having two combined, and the best performance is achieved with the three (DNA, RNA, and protein) modalities together (Table 2). This is true for both Spearman correlation and $R^2$ metrics, with stronger improvement for the latter reflecting a more accurate prediction of the actual values of expression and not just the ranking of transcripts. This is a strong demonstration that our model can aggregate information across modalities to improve performance on this isoform expression task. In addition, we observe increased performance by using DNA together with RNA compared with DNA and protein information. This can be related to the strong importance of the UTR regions of the RNA sequence in the regulation of its degradation and stability [61], which affect its final expression level in the cells, that are not captured at the protein level.

## 5.2 Enformer as DNA encoder module to obtain transfer between expression prediction tasks

**Experiment** To showcase the flexibility of IsoFormer towards different modality-specific encoders, we tested replacing NT by the Enformer model [14] as DNA encoder. Enformer has been trained over gene-level expression data obtained from CAGE assays (one value of expression per gene per tissue), while our model is trained to predict RNA transcript expression data obtained from bulk RNA-seq assay (one value of expression for each isoform of a given gene per tissue) and therefore represents a different challenge that cannot be tackled from the DNA sequence alone. Still, as the Enformer has been trained to predict gene-level expression across tissues directly from DNA sequences, a related task to predicting RNA isoform expression, one might expect to obtain transfer by using it as DNA encoder. We give full details of this experiment in Appendix D.

**Results** We obtained superior performance using the Enformer instead of NT as a pre-trained DNA encoder both as DNA-only but also when we combined with the RNA and protein encoders (Table 3). Importantly, also with the Enformer our framework benefits from bridging modalities. This improved

Table 3: Comparison of Enformer and NT DNA encoders used in IsoFormer. $R^2$ and Spearman correlation across tissues on the transcript isoform expression prediction task. Standard deviation across 5 seeds is reported.

| Model | $R^2$ | Spearman |
|-------|-------|----------|
| Enformer | $0.21 \pm 0.01$ | $0.46 \pm 0.00$ |
| IsoFormer (NT) | $0.43 \pm 0.01$ | $0.65 \pm 0.01$ |
| IsoFormer (Enformer) | $\mathbf{0.53 \pm 0.01}$ | $\mathbf{0.72 \pm 0.00}$ |
| IsoFormer (Borzoi) | $0.48 \pm 0.01$ | $0.69 \pm 0.00$ |

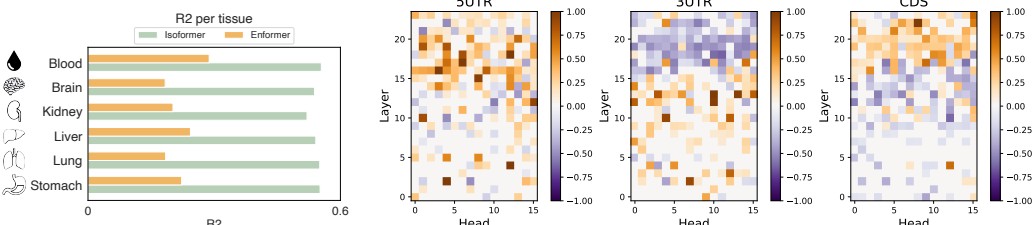

Figure 3: **Left:** Performance of IsoFormer and Enformer [14] per tissue on a selected subset of tissues. **Right:** Changes in attention in the RNA encoder during fine-tuning. These scores are reported for three genomics elements of interest for all heads and layers of the RNA encoder.

performance can be explained by the fact that Enformer has been pre-trained on the related task of gene expression prediction and thus its embeddings are better aligned with the isoform prediction task. Moreover, the Enformer is a model that can handle sequences of large context (up to 196k nucleotides), enabling it to capture long-range dependencies, known to be relevant for expression. We also tried the Borzoi model [] as an enocder for DNA model. Borzoi is a model pre-trained on sequences up to 512kb nucleotides to predict RNA-seq coverage which directly relates to RNA transcript isoform exression. These results demonstrate that our multi-modal framework can be improved by leveraging more domain-specific encoders. As our best model for isoform prediction is achieved using the Enformer as DNA encoder, we will use it as DNA encoder by default for all the following experiments. We make the weights of this IsoFormer model available on HuggingFace[5].

**Interpretation** We report the performance of IsoFormer across selected tissues in Fig. 3-left (results across all tissues are presented in Appendix D). IsoFormer obtains similar performance across tissues despite tissues having different distributions of expression levels. To gain additional insights about the representations learned by IsoFormer, we analysed the attention layers inside the RNA encoder as it is the one providing stronger improvement on this task. Specifically, we compared how the attention distribution within each layer and head changes when we finetune the RNA encoder alone versus finetuning IsoFormer altogether. We report changes in attention scores at each layer and head of the RNA encoder for three genomics elements known to have a strong effect on the isoform splicing and gene expression processes, namely the 3UTR, 5UTR and CDS sequence, see Fig. 3-right (additional details on these scores are in Appendix Section C.1). The results show that, when finetuning using the three modalities, different layers specialize to capture specific features relative to isoform splicing and expression. Notably, the middle set of layers put higher attention weights to 3UTR regions whereas the top layers of NT attributes higher attention weights on CDS and 5UTR. We assume that this RNA encoder specialization during finetuning is key to achieve a strong representation towards the prediction of its tissue-specific expression.

### 5.3 Ablation studies on the aggregation strategy

**Experiment** We compared the IsoFormer's aggregation module with alternative strategies from recent multi-modal literature (Fig. 4). Inspired by recent vision-text models [20, 52, 62, 63], we considered these three approaches: (i) *Perceiver Resampler* [50]: a variant of our cross-attention method using a Perceiver Resampler module. This block learns a fixed number of tokens for each modality, thus reducing the cost of the subsequent cross-attention layer. (ii) *Linear Projection*: a strategy that linearly projects the embeddings of the three modalities into a common representation space and concatenates all tokens. This concatenated sequence is fed to a Perceiver Resampler to

---

[5]https://huggingface.co/InstaDeepAI/isoformer.

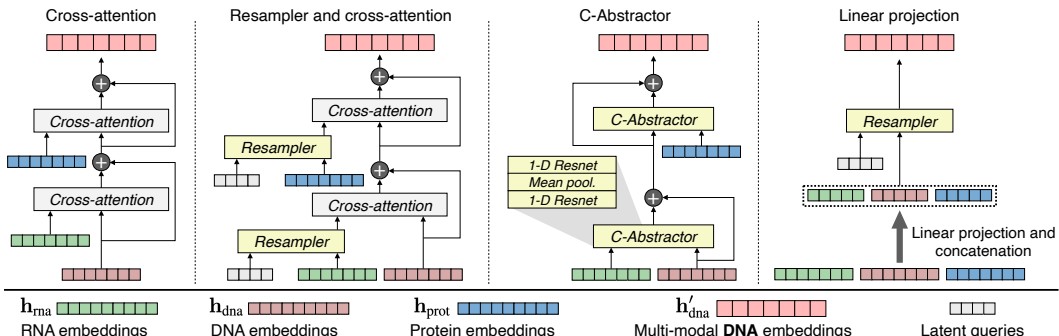

Figure 4: Different aggregation strategies compared during the ablation studies. The figures show the specific case for obtaining multi-modal DNA embeddings $\mathbf{h}'_{dna}$; the same structure is used to obtain multi-modal RNA and protein embeddings ($\mathbf{h}'_{rna}$ and $\mathbf{h}'_{prot}$, respectively). In all cases, the *Resampler* module is a *Perceiver Resampler* [50] and the *Mean pooling* block is the *Adaptive mean pooling* operator used in [20].

learn a fixed number of tokens which are then used in the head of the model. (iii) *C-Abstractor* a 1-dimensional version of the C-abstractor architecture that provides a compromise between flexibility –choosing an arbitrary number of resampled tokens– and locality preservation [20].

Table 4: Ablation study of different aggregation strategies when considering the three modalities together. All experiments were run using Enformer as the DNA encoder. PR = Perceiver Resampler.

| Aggregation Method | $R^2$ | Spearman |
|---|---|---|
| Ours | $0.53 \pm 0.01$ | $0.72 \pm 0.00$ |
| PR + Cross-Attn | $0.49 \pm 0.04$ | $0.69 \pm 0.02$ |
| Linear Proj. + PR | $0.49 \pm 0.02$ | $0.69 \pm 0.01$ |
| C-Abstractor | $0.53 \pm 0.01$ | $0.72 \pm 0.01$ |

**Results**   We performed hyperparameters search with the same budget used for the IsoFormer aggregation module for all ablations methods. We report in Table 4 the results obtained with the best set of hyperparameters for each method. We observe that, using an additional step of Perceiver Resampler always performs worse than only cross-attention –even after optimizing hyperparameters. Similarly, C-Abstractor does not confer any benefit over cross-attention; therefore we consider this latter strategy as the optimal aggregation strategy for our method. One advantage of the cross-attention mechanism we use is its interpretability, since it helps understanding which regions of the different modalities are being leveraged to make predictions. These conclusions align with recent multi-modal studies for other modalities [64].

## 5.4   IsoFormer leverages knowledge of pre-trained encoders

**Experiment**   We showed in previous sections that IsoFormer is able to aggregate information from different biological sequence modalities to formulate high-quality predictions. Here, we investigated to which extent IsoFormer's performance can be attributed to the transfer from each modality-specific encoders' pre-training. We conduct extensive experiment where we compared the performance of IsoFormer trained using all three encoders pre-trained on their respective domains with different IsoFormer model variants with all possible combinations of pre-trained / non-pretrained modality-specific encoder.

**Results**   Our results demonstrate that using pre-trained encoders confers a substantial advantage to IsoFormer, as the $R^2$ is substantially larger (0.53) when compared with IsoFormer with none of the encoders pre-trained (0.10; Table 5). This demonstrates that IsoFormer is leveraging the knowledge acquired by each foundation model in the respective domains. However, we observed that when we randomly initialized only the DNA or RNA encoders, the drop in performance is smaller (0.41 for DNA and 0.48 for RNA). This suggests that IsoFormer not only leverages intra-modalities pre-training but also inter-modalities transfer. Altogether, these results underpin our approach of relying on initializing IsoFormer with pre-trained encoders, as the information learned during pre-training is transferred and leveraged when considering multi-modal tasks.

Table 5: Comparing the use of pre-trained and non-pre-trained encoders within IsoFormer. For this set of experiments the considered encoders are the Enformer for DNA, NT for RNA and ESM for proteins. ✓ indicates the use of a pre-trained encoder whereas ✗ indicates the encoder is trained from scratch (random initialization).

| DNA | RNA | Protein | $R^2$ | Spearman |
|:---:|:---:|:---:|:---:|:---:|
| ✗ | ✗ | ✗ | $0.10 \pm 0.03$ | $0.31 \pm 0.01$ |
| ✓ | ✗ | ✗ | $0.45 \pm 0.01$ | $0.67 \pm 0.00$ |
| ✗ | ✓ | ✗ | $0.39 \pm 0.01$ | $0.61 \pm 0.00$ |
| ✗ | ✗ | ✓ | $0.34 \pm 0.01$ | $0.59 \pm 0.01$ |
| ✓ | ✓ | ✗ | $0.52 \pm 0.01$ | $0.71 \pm 0.00$ |
| ✓ | ✗ | ✓ | $0.48 \pm 0.01$ | $0.69 \pm 0.00$ |
| ✗ | ✓ | ✓ | $0.41 \pm 0.01$ | $0.64 \pm 0.01$ |
| ✓ | ✓ | ✓ | $0.53 \pm 0.01$ | $0.72 \pm 0.00$ |

## 6   Conclusion

IsoFormer is the first model designed for multi-modal biological sequence modeling connecting DNA, RNA, and protein sequences. IsoFormer achieves state-of-the-art results by effectively leveraging and transferring knowledge from pre-trained DNA-, RNA-, and protein-specific encoders on one of the significant multi-modal problems in genomics: RNA *transcript isoform expression prediction*. As part of our efforts, we are are open-sourcing our model, and hope IsoFormer paves the way to new milestones in building multi-modal models for biology.

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

# A  Dataset details

We based our dataset on the Genotype-Tissue Expression (GTEx) portal. Specifically, we use the 8th release of the *Transcript TPMs* table[6]. This table is made of expression measurements of $170k$ transcripts across 30 different tissues from $5,000$ individuals. As the goal is to build a general model for expression prediction from biological sequences, we averaged measurements across individuals to get an average expression value for each transcript in each tissue.

We mapped each transcript to its corresponding gene and protein using the Ensembl[7] database. Using this database, we were able to retrieve associated DNA, RNA, and protein sequences. For DNA sequences, we used the latest release of the human reference genome *GRCh38*[8].

To summarize, the steps to re-create our training dataset are the following:

1. Download *Transcript TPMs* table from GTEx portal[9]

2. Compute average measurement per transcript ID and tissue

3. Map RNA transcript isoform ID to its corresponding RNA sequence using Ensembl

4. Get associated protein isoform using Ensembl

5. Get chromosome and transcription start site position on the DNA sequence

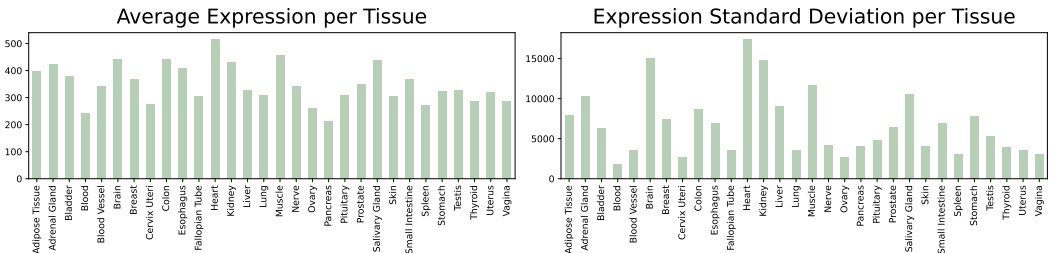

Figure 5: Average and standard deviation of expression values across transcripts per tissue.

# B  Training and architecture details

All experiments were carried out with 5 seeds on 4 A100 GPUs (80GB RAM). Depending on the model, a training run lasts between 1 and 5 hours. We provide model hyper-parameters in Table 6 and the hyper-parameters of the encoders in Table 7.

In order to stabilize our training procedure, expression values were processed in two steps: first, we use a log-transform ($val = \log(1 + val)$) to minimize the effect of outliers; secondly, we normalized expression values per tissue as it has been shown to stabilize training when using mean squared error loss.

# C  Experimental details

## C.1  Attention maps analysis

Section 5.2 introduces significant results on IsoFormer's ability to leverage pre-trained modality-specific encoders. When finetuning the pre-trained encoders together, there is specialization on specific elements of the sequence, as observed in the RNA attention maps in Figure 3-right. These maps have been obtained through the following process:

---

[6]`https://www.gtexportal.org/home/downloads/adult-gtex/bulk_tissue_expression`
[7]`https://www.ensembl.org/index.html`
[8]`https://www.ncbi.nlm.nih.gov/datasets/genome/GCF_000001405.26/`
[9]`https://www.gtexportal.org/home/downloads/adult-gtex/bulk_tissue_expression`

Table 6: Model hyper-parameters

| Hyper-parameter | Value |
| --- | --- |
| Cross-Attn: number of heads | 8 |
| Perceiver Resampler: number of layers | 1 |
| Perceiver Resampler: number of resampled tokens | 8 |
| C-abstractor: kernel size | 3 |
| C-abstractor: number of residual layers | 2 |
| C-abstractor: number of resampled tokens | 8 |
| Maximum number of nucleotides in DNA sequences (Enformer) | 196,608 |
| Maximum number of nucleotides in DNA sequences (NT-v2) | 12,288 |
| Maximum number of nucleotides in RNA sequences | 12,288 |
| Maximum number of amino-acids in protein sequences | 1,200 |

Table 7: Encoder hyper-parameters

| Hyper-parameter | Value |
| --- | --- |
| NT: maximum number of tokens | 2,048 |
| NT: number of attention heads | 16 |
| NT: embedding dimension | 768 |
| NT: number of layers | 24 |
| NT: activation | swish |
| Enformer: number of parameters | 110M |
| Enformer: embedding dimension | 1,536 |
| Enformer: number of Transformer layers | 8 |
| *ESM-2-150M*: number of attention heads | 20 |
| *ESM-2-150M*: embedding dimension | 640 |
| *ESM-2-150M*: number of layers | 30 |

1. Focusing on the pre-trained RNA encoder, we take the attention weights for each layer and head after running the IsoFormer finetuning process. For each layer and head, we compute how much attention (percentage) is directed towards a specific region of interest of the RNA sequence. The regions we consider are 3UTR, 5UTR, and CDS, and we use the following equation to compute the exact ratio of attention:

$$\rho(f) = \frac{1}{\mathbf{X}_{\text{rna}}} \sum_{x \in \mathbf{X}_{\text{rna}}} \frac{\sum_i \sum_j f(i) \mathbf{1}(\alpha(i,j) > \mu)}{\sum_i \sum_j \mathbf{1}(\alpha(i,j) > \mu)} \tag{6}$$

where $\mathbf{X}_{\text{rna}}$ is the set of RNA sequences in the test set, $\alpha(i,j)$ is the attention coefficient between tokens $i$ and $j$, $f(i)$ is an auxiliary function that equals 1 if token $i$ belongs to the region of interest in the sequence (e.g. 3UTR), and $\mu$ is a threshold value (we choose 0.01). We denote these attention maps by $\rho_{\text{IF}}$.

2. We repeat this process for the finetuning run in which only the RNA encoder is used (RNA only in Table 2). We denote these attention maps by $\rho_{\text{NT}}$.

3. For each layer and head, we compute the ratio $\Delta\rho = (\rho_{\text{IF}} - \rho_{\text{NT}})/\rho_{\text{NT}}$. For simplicity, we cap these values to 1 (i.e., attention rate is doubled in the IsoFormer case compared to finetuning a RNA encoder alone).

4. In addition, for both $\rho_{\text{IF}}$ and $\rho_{\text{NT}}$, we can consider the samples in the test set $\mathbf{X}_{\text{rna}}$ and build a distribution of attention rates per element of interest. Comparing both distributions (the one coming from $\rho_{\text{IF}}$ and the one coming from $\rho_{\text{NT}}$), we can carry out a t-test per layer and head. We follow these t-tests and select the combinations of layers and heads in which there are statistically significant differences (i.e., $p < 0.05$) between $\rho_{\text{IF}}$ and $\rho_{\text{NT}}$.

5. Figure 3-right shows the ratio $\Delta\rho$ for those pairs of layers and heads in which statistically significant differences are observed. The rest is set to zero.

# D  Additional results

## D.1  Full results over every tissue

Figure 6 shows the performance of IsoFormer using Enformer as DNA encoder for each of the 30 tissues. IsoFormer outperforms the DNA-only Enformer model on every tissue.

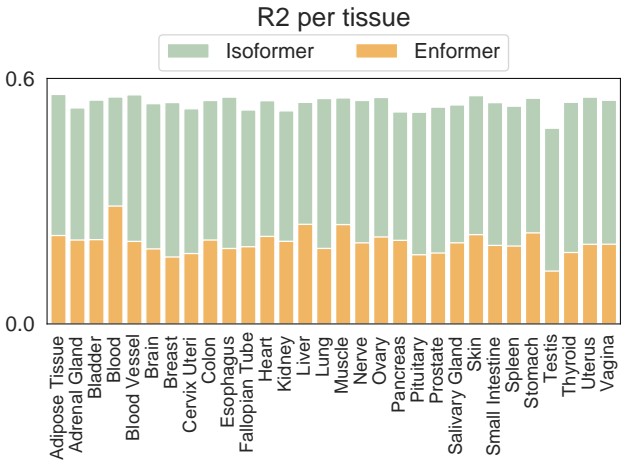

Figure 6: Performance of IsoFormer and Enformer [14] per tissue on all tissues.

## D.2 Full ablation studies on modalities

Table 8: Performance of different variants of IsoFormer for the prediction of transcript isoform expression using three different DNA encoders: Nucleotide Transformer, Enformer, and Borzoi (input sequence length in parentheses). $R^2$ and Spearman correlation across tissues for 5 different random seeds is reported. NT is used as RNA encoder while ESM is used to process protein sequences.

| Base DNA Model | NT (12k) | | Enformer (196k) | | Borzoi (512k) | |
|---|---|---|---|---|---|---|
| | $R^2$ | Spearman | $R^2$ | Spearman | $R^2$ | Spearman |
| DNA Only | $0.13 \pm 0.02$ | $0.43 \pm 0.01$ | $0.21 \pm 0.02$ | $0.46 \pm 0.00$ | $0.12 \pm 0.02$ | $0.35 \pm 0.01$ |
| RNA Only | $0.36 \pm 0.03$ | $0.61 \pm 0.01$ | $0.36 \pm 0.03$ | $0.61 \pm 0.01$ | $0.36 \pm 0.03$ | $0.61 \pm 0.01$ |
| Protein Only | $0.20 \pm 0.01$ | $0.46 \pm 0.01$ | $0.20 \pm 0.01$ | $0.46 \pm 0.01$ | $0.20 \pm 0.01$ | $0.46 \pm 0.01$ |
| RNA + Protein | $0.40 \pm 0.01$ | $0.63 \pm 0.01$ | $0.40 \pm 0.01$ | $0.63 \pm 0.01$ | $0.40 \pm 0.01$ | $0.63 \pm 0.01$ |
| DNA + Protein | $0.28 \pm 0.01$ | $0.52 \pm 0.01$ | $0.39 \pm 0.01$ | $0.61 \pm 0.01$ | $0.30 \pm 0.02$ | $0.56 \pm 0.01$ |
| DNA + RNA | $0.39 \pm 0.01$ | $0.64 \pm 0.01$ | $0.52 \pm 0.01$ | $0.72 \pm 0.01$ | $0.47 \pm 0.01$ | $0.69 \pm 0.00$ |
| DNA + RNA + Prot. | $0.43 \pm 0.01$ | $0.65 \pm 0.01$ | $0.53 \pm 0.01$ | $0.71 \pm 0.00$ | $0.48 \pm 0.01$ | $0.69 \pm 0.00$ |

