# OpenReview forum: "Multi-modal Transfer Learning between Biological Foundation Models"
_NeurIPS.cc/2024/Conference — NeurIPS 2024 poster_

### Official Review · Reviewer_Z3Wd · 2024-07-05

**Soundness:** 3
**Presentation:** 3
**Contribution:** 2
**Rating:** 6
**Confidence:** 3

**Summary:**

The paper introduces a novel multi-modal model, IsoFormer, designed to integrate DNA, RNA, and protein sequences for predicting RNA transcript isoform expression across different tissues. It utilizes pre-trained modality-specific encoders to generate embeddings that are then combined using a sophisticated aggregation method. The model demonstrates significant improvements in prediction accuracy compared to single-modality approaches.

Contriburion:

1. Developed the first general-purpose multi-modal model integrating DNA, RNA, and protein sequences.

2. Demonstrated successful application of transfer learning from modality-specific encoders.

3. Provided a new robust framework for advancing the prediction of RNA transcript isoform expression.

**Strengths:**

1. Innovative Integration of Modalities: The paper presents the first attempt to integrate three biological sequence modalities (DNA, RNA, and proteins) in a unified model, providing a comprehensive approach reflective of natural biological processes.

2. Effective Transfer Learning: IsoFormer effectively leverages pre-trained encoders to enhance its predictive power, benefiting from both intra-modal and inter-modal transfer learning.

3. Robust Evaluation: Experiments demonstrate the model's capability, outperforming existing methods in predicting transcript isoform expression, which is a challenging task due to its multi-modal nature.

**Weaknesses:**

1. Complexity and Computation: The model's complexity and the computational demands might limit its accessibility and use, particularly in environments with restricted resources.

2. More Comprehensive Evaluation for PLM's representation learning capability would make this paper better.

**Questions:**

(1)	For Tab 5., wonder what’s the performance for “DNA and RNA encoder not pre-trained”

(2)	Could authors provide evaluation results for DNA, RNA and protein encoder separately on their own popular benchmarking tasks? Ideally, these models should improve on those downstream tasks as well. Would give 7’ or even 8’ if authors conduct these experiments. It could be a great and exciting work in this AI4Bio field.

**Limitations:**

1. Data Requirements: The effectiveness of the model is contingent on the availability of comprehensive and high-quality multi-modal datasets.

2. Generalizability: While promising, the results are primarily validated on specific types of gene expression data, and its performance across broader biological applications remains to be fully assessed.

---

> ### Author Rebuttal · Authors · 2024-08-06
>
> We thank the reviewer for the insightful feedback and positive comments of our work.
>
> > For Tab 5., wonder what’s the performance for “DNA and RNA encoder not pre-trained”
>
> We have now completed this ablation study with full evaluation of the effect of pre-training for each encoder (detailed in the attached PDF, Table 2). Interestingly, we observe in every setting that using a pre-trained encoder against its randomly initialized counterpart consistently leads to improved performance. This further demonstrates that IsoFormer leverages modality-specific pre-training.
>
> > Could authors provide evaluation results for DNA, RNA and protein encoder separately on their own popular benchmarking tasks? Ideally, these models should improve on those downstream tasks as well. Would give 7’ or even 8’ if authors conduct these experiments. It could be a great and exciting work in this AI4Bio field.
>
> We evaluated the downstream performance of the Nucleotide Transformer model before and after fine-tuning it within the Isoformer on the isoform prediction task (when considering it as a base DNA model). We selected the prediction of gene expression measured by bulkRNA-seq from Chia Hsiang et al. [1] as it aligns with the fine-tuning task of the IsoFormer while relying on a different data source. We show a clear increase in performance of the NT model before and after fine-tuning within the IsoFormer framework (see attached PDF, Table 3). We will add these results to the paper. This highlights the potential of our approach to jointly pre-train models for different modalities. We will discuss in the conclusion section how this could be extended to a larger set of downstream tasks across modalities, which will probably require to expend the tasks and datasets used to train the IsoFormer.
>
> > Data Requirements: The effectiveness of the model is contingent on the availability of comprehensive and high-quality multi-modal datasets.
>
> The reviewer brings up one of the key motivating aspects of our work. One of the biggest bottlenecks in the development of models integrating multiple biological sequences has been the lack of available matched data between DNA, RNA, and proteins. These multi-modal datasets are becoming more available but the amount of data is not enough to train models from scratch (as we demonstrate in Table 5 of the paper). Our approach bridges this gap by leveraging uni-modal encoders that have been trained on high-quality uni-modal datasets and achieving transfer learning across modalities. Our work demonstrates that this limitation can be overcome.
>
> [1] Kao, Chia Hsiang, et al. "Advancing dna language models: The genomics long-range benchmark." ICLR 2024 Workshop on Machine Learning for Genomics Explorations. 2024.

---

> > ### Comment · Reviewer_Z3Wd · 2024-08-10
> >
> > Hi thanks for all the efforts and detailed response. For my question (2), we refer to benchmarking tasks used in protein language models, RNA language models, and DNA language models, not only on transcript isoform expression prediction task. Not sure if the remaining time would be enough. Sorry that I didn't response promptly because of work during weekdays.

---

> > > ### Author Response · Authors · 2024-08-13
> > > **Clarification on the additional gene expression task**
> > >
> > > For clarification, the additional experiment is the prediction of gene expression measured by bulkRNA-seq from Chia Hsiang et al. [1], which is different from the isoform expression prediction task that was used to train IsoFormer. In the transcript isoform expression prediction task, the goal is to predict the expression of each RNA transcript isoform of a given gene, whereas in the gene expression prediction task, the goal is to predict the expression of a gene from its DNA sequence - and this is one of the key benchmarking tasks for DNA language models nowadays. Those tasks are different but correlated, hence we chose it as we expect transferability between them. Our experiment demonstrates that training Nucleotide Transformer as a DNA encoder within our multi-modal transcript expression framework increases its performance on this gene expression task.
> > >
> > > We did not have the time to run extensive experiments on existing benchmarks for protein and RNA language models before the end of discussion period but we agree that those would be valuable insights. This is an interesting step for future iterations of this work that we will add to the conclusion of the paper.
> > >
> > > [1] Kao, Chia Hsiang, et al. "Advancing dna language models: The genomics long-range benchmark." ICLR 2024 Workshop on Machine Learning for Genomics Explorations. 2024.

---

### Official Review · Reviewer_pNzb · 2024-07-08

**Soundness:** 2
**Presentation:** 2
**Contribution:** 2
**Rating:** 5
**Confidence:** 4

**Summary:**

The paper introduces a new framework for the multi-modality pretrain model according to the Central dogma of biology. The method encode DNA, protein and RNA at the same time. The proposed method can transfer knowledge from the encoders pretraining and modalities.

**Strengths:**

The paper is well-organized and easy to follow
The authors have proved that the multi-modality of single cell data can help model predictions.

**Weaknesses:**

Lack of experiments. 1. More ablation studies should be conducted about removing different modalities of the model in Table 2 ( e.g. we observe only RNA can achieve a high performance, what about protein+RNA? ).

 2. More dataset details should be included. The split of training/validation/test sets is not clear. If the authors do the experiments on the same dataset, they should split the dataset according to the tissues to validate the transferability of the proposed method.

**Questions:**

The authors should include more motivation about the proposed method. For example, why can the changed DNA influence the RNA seq? Why not directly predict RNA expression from RNA seq?

**Limitations:**

The biological system is more complex. The proposed method only include the direct map from DNA to RNA. However, in real world, RNA can effect the expression of DNAs. More details should be discuss in the future work.

---

> ### Author Rebuttal · Authors · 2024-08-06
>
> We appreciate this reviewer's comments and suggestions that will improve the revised manuscript.
>
> > More ablation studies should be conducted about removing different modalities of the model in Table 2 ( e.g. we observe only RNA can achieve a high performance, what about protein+RNA? ).
>
> We have now performed the suggested, and missing, ablation studies for NT, Enformer and Borzoi (added now) models (see attached PDF, Table 1). We now can compare the results of using any combination of one, two, or three modalities with three different DNA encoders. These results reinforce the advantage of combining multiple modalities to solve the isoform expression prediction task. The new version of the paper will incorporate these results and their appropriate interpretation. Many thanks for the suggestion.
>
> > More dataset details should be included. The split of training/validation/test sets is not clear. If the authors do the experiments on the same dataset, they should split the dataset according to the tissues to validate the transferability of the proposed method.
>
> Regarding the data split in training/validation/test sets, we will provide more details on the methods. Briefly, we have followed standard approaches for sequence-based models where different chromosomes are held out for the different sets (chromosomes 1-19 for training, 20 for validation and 21-22 for test). This ensures no data leakage due to overlapping or recombinant sequences. Since the only input are DNA/RNA/AA sequences, we do not expect transferability between tissues and can only do tissue-specific predictions for the tissues used during training.
>
> > The authors should include more motivation about the proposed method. For example, why can the changed DNA influence the RNA seq? Why not directly predict RNA expression from RNA seq?
>
> We will also include more motivation about the proposed method. For further clarification, here we are predicting isoform expression which is measured by RNA-seq. Thus, it would be trivial to predict it from RNA-seq and the main challenge is to predict it from DNA sequence. Transcript isoform expression is influenced by DNA regulatory elements in addition to RNA regulatory elements, and that is the main motivation for combining multiple encoders to improve performance on this task.
>
> > The biological system is more complex. The proposed method only includes the direct map from DNA to RNA. However, in the real world, RNA can affect the expression of DNAs. More details should be discussed in the future work.
>
> We will also improve the discussion of the complexity of gene regulation and the system. Indeed, the fact that RNA can also affect in turn the expression of genes from DNA is one of the main motivations to mix DNA and RNA sequences in our approach, modeling interactions between biological modalities based on their sequence alone.
>
> We hope our revisions address your concerns. We would be happy to expand on any of these points throughout the discussion period if further clarification is needed.

---

> > ### Comment · Reviewer_pNzb · 2024-08-13
> >
> > Thanks for your response. I have raised my score. Hope your next version can be more clear

---

### Official Review · Reviewer_p67W · 2024-07-15

**Soundness:** 3
**Presentation:** 2
**Contribution:** 2
**Rating:** 6
**Confidence:** 5

**Summary:**

The paper models isoform relative abundance across tissues with a multimodal approach based on 3 pretrained encoders for DNA, RNA, and AA sequences.  DNA encoder uses a sequence centered on the gene’s TSS, RNA encoder uses the known isoform sequence from RNAseq and the protein encoder uses corresponding AA sequence.  They perform multiple ablations on the utility of having all 3 separate encoders and, given separate encoders, how to aggregate them into a single isoform specific embedding/prediction, and look at attention layers of RNA module to find biologically meaningful regions of attention.

**Strengths:**

Isoform level analysis using 3 separate pretrained encoders for DNA, RNA, and AA sequences is a good strategy.  The authors provide useful ablations on the utility of the multi modal approach and on modern strategies for combining those into a single embedding.  Looking for biolgoically meaningful interpretations of attention layers is useful.

**Weaknesses:**

I don’t think the authors can claim this is the first attempt to combine DNA, RNA, and AA modalities with techniques from NLP.   See the recent Evo work here https://www.biorxiv.org/content/10.1101/2024.02.27.582234v2 .  While they evaluate their performance against Enformer, that’s a large part of their own model.  So the evaluations have an intramural feel to them.  It’d be interesting to see how their strategy compares to other multi modal models such as Evo, and more RNA centric work like Borzoi, which looks at a more fine grained look of variant effects on the DNA to RNA relationship.  Looking at average isoform abundance across individuals is all well and good, but GTEx also has individual genomes, and genomic variation across individuals will also of course affect splicing patterns and which isoforms come from what individuals.

**Questions:**

Some comments and questions:

Centering on TSS will only capture regulatory elements within the chosen sequence length.  There could be distal or trans CREs outside

GTEx database has less than 1000 donors, confirming the 5000 individuals claim?

Looking at equation 5 in 4.4, how does f_psi depend on the tissue T? Is it a separate head per tissue, or are you predicting the vector of isoform abundance across tissues with one pass?  And is f_theta and f_phi the same f but different weights as f_psi?  Where does the summation over i take place?

5.1 does ablations with one DNA encoder, then 5.2 shows superior performance with Enformer as the DNA encoder.  So the ablations in 5.1 may not be accurate with respect to this new encoder.  It also begs the question of how would the Enformer do as the RNA encoder as well.

How are RNA and protein sequence lengths handled when they’re longer than the model input sequence size?

**Limitations:**

The authors should be more explicit about the limitation of using reference genome and known isoform sequences and how this kind of sweeps splicing as a function of dna sequence under the rug.

---

> ### Author Rebuttal · Authors · 2024-08-06
>
> Many thanks for the constructive comments and positive assessment of our work.
>
> > I don’t think the authors can claim this is the first attempt to combine DNA, RNA, and AA modalities with techniques from NLP. See the recent Evo work
>
> Evo is a model based solely on DNA sequences and has been applied to RNA and protein modalities using their respective DNA sequences, a technique used in recent models [1,2]. Evo was trained on prokaryotic data, while ours uses eukaryotic data. For eukaryotes, models trained on genomes show limitations due to data distribution differences arising from the presence of intronic regions [2]. Our approach uses different vocabularies per modality and models pre-trained on corresponding datasets. Therefore, Evo and our models are not directly comparable and have different strengths. Ours is the first to combine DNA, RNA, and AA modalities with their respective alphabets.
>
> > It’d be interesting to see how their strategy compares to other multi modal models such as Evo, and more RNA centric work like Borzoi
>
> We have now compared our multi-modal approach using Enformer and Borzoi in addition to NT. Interestingly, using Borzoi as DNA encoder leads to the same conclusions on the positive effect of aggregating more modalities (see attached PDF, Table 1), maintaining the ranking between the different ablations. This observation supports the robustness of our approach. Additionally, Borzoi outperformed NT as a DNA encoder, which is expected since Borzoi is tailored for expression tasks. However, despite the Borzoi authors reporting similar performance to Enformer in modeling gene expression, we observed slightly lower performance for Borzoi.
>
> >  Looking at average isoform abundance across individuals is all well and good, but GTEx also has individual genomes, and genomic variation across individuals will also of course affect splicing patterns and which isoforms come from what individuals.
>
> > The authors should be more explicit about the limitation of using reference genome and known isoform sequences and how this kind of sweeps splicing as a function of dna sequence under the rug.
>
> Genomic variation across individuals will does affect splicing patterns, we are not accounting for that for now. GTEx also has individual genome data but we do not have access to it yet. We will highlight this point in the paper and propose as future experiments to incorporate genetic variants. Still, this should not impact the comparison between using the different modalities alone or in combination, as in all cases they use the reference genome sequences.
>
> > Centering on TSS will only capture regulatory elements within the chosen sequence length. There could be distal or trans CREs outside.
>
> It is true that centering on TSS will only capture regulatory elements within the chosen sequence length. But when using Enformer we use a context window of ~190kb, thus capturing most distal enhancers, being the state-of-the-art method to capture such long interactions. Similarly, when using Borzoi in our new experiments, we use a context window of ~512kb.
>
> > GTEx database has less than 1000 donors, confirming the 5000 individuals claim?
>
> We indeed had a typo and will correct it in the revised version, thanks for pointing this out.
>
> > Looking at equation 5 in 4.4, how does f_psi depend on the tissue T? Is it a separate head per tissue, or are you predicting the vector of isoform abundance across tissues with one pass? And is f_theta and f_phi the same f but different weights as f_psi? Where does the summation over i take place?
>
> f_psi returns indeed the vector of isoform abundance across tissues in one pass, returning a vector of n_t floating numbers where n_t is the number of tissues. The reviewer is right, the notation f is confusing. f_theta, f_phi and f_psi are different functions parametrized with different weights; we'll rename them f^{encoders}_theta, f^{aggregation}_phi and f^{expression}_psi to improve clarity.
>
> For the sake of simplicity, we removed the summation over batches of data in the loss function. The summation over i takes place implicitly as we sample randomly isoforms and their expression in the dataset. We do not necessarily have all isoforms from the same gene within the same batch. We will add a sentence in the text to clarify this.
>
> > 5.1 does ablations with one DNA encoder, then 5.2 shows superior performance with Enformer as the DNA encoder. So the ablations in 5.1 may not be accurate with respect to this new encoder. It also begs the question of how would the Enformer do as the RNA encoder as well.
>
> We have also now performed all encoder ablations for NT, Enformer, and Borzoi models, providing more insights about the benefits of combining multiple encoders (see attached PDF, Table 1). Many thanks for the suggestion.
>
> Regarding using Enformer as the RNA encoder, since it is long-range but most RNAs are shorter than 10kb, we favored the use of a more general RNA encoder. This could be done as future work, or rather using recently published RNA pre-trained models.
>
> > How are RNA and protein sequence lengths handled when they’re longer than the model input sequence size?
>
> We will clarify in the revised version how we handle RNA and protein sequences that exceed the model's input size. Protein sequences were all within the model's maximum input length, so no additional processing was required. RNA transcripts longer than 12kb were left-cropped to 12kb to conserve the 3’UTR regions, which are crucial for mRNA stability and polyadenylation, impacting isoform abundance. This adjustment affected only a small portion of the dataset. We will include the sequence length distributions in the paper's supplementary material.
>
> [1] Richard, G., et al. "ChatNT: A Multimodal Conversational Agent for DNA, RNA and Protein Tasks." bioRxiv (2024): 2024-04.
>
> [2] Outeiral, C., and C. M. Deane. "Codon language embeddings provide strong signals for protein engineering. bioRxiv." preprint (2022).

---

> > ### Comment · Reviewer_p67W · 2024-08-11
> >
> > thank you for addressing my questions.  i think those corrections and clarifications will benefit the paper’s presentation.

---

### Author Rebuttal · Authors · 2024-08-06

We would like to thank the reviewers for their time reading our manuscript and for providing constructive feedback in their reviews.

We are glad the reviewers value positively our approach to combine different biological modalities together and emphasize that the experimental results support the paper’s contributions. We also acknowledge certain areas of the manuscript can be strengthened and are grateful to the reviewers for providing improvement suggestions on those areas. Below, we summarize the main concerns that have been raised during the review and the changes we have made to address them. Concrete concerns and questions are further discussed in the reviewer-specific responses.

**Full enumeration of modality combinations and use of RNA-centric models like Borzoi**

Reviewers suggested expanding the results presented in Tables 2 and 3 of the paper to include 1) any combination of one, two, or three modalities, and 2) the use of Borzoi as one of the encoders. We agree with the suggestion and present exhaustive results in Table 1 of the attached PDF. These results reinforce the advantage of combining multiple modalities to solve the isoform expression prediction task. In addition, we can observe that when using Borzoi as a DNA encoder the ranking between the different ablations is maintained. We will include this updated table in the final version of the paper.

**Full enumeration of using the pre-trained and non-pre-trained versions of each encoder**

Table 5 of our paper compares different cases in which only a subset of the encoders is pre-trained. This table demonstrates the effectiveness of transfer learning from pre-trained uni-modal encoders to achieve a strong performance in the multi-modal task. Reviewers suggested extending this table to include the 8 possible combinations of pre-trained vs. non-pretrained. We provide the complete ablation in Table 2 in the attached PDF. The extended results show in every setting that using a pre-trained encoder against its randomly initialized counterpart consistently leads to improved performance. We will add this table to the final version of the paper.

**Clarification of the data processing approach**

The reviewers have inquired about our specific use of the GTEx data. First, an interesting point has been made on using reference genome vs. individual genomes. We have opted to use the reference genome in our work because of the availability of the data. While we agree it would be interesting to use individual genome data, we believe that the reference genome is already a strong support for proving the main goal of the paper: combining multiple biological sequence modalities together to obtain superior performance in multi-modal tasks. We will discuss the use of individual genome data as future work in the final version of the paper.

Second, reviewers have asked to provide further details on the train/validation/test splits. In our work we have created these splits based on chromosomes, which is a standard approach in sequence-based models and ensures there is no data leakage across splits. We add the specific splits in our response and in the final version of the paper. Finally, we have also addressed other specific points about the dataset such as the treatment of long sequences (we crop them, not necessary when using Enformer) and the motivation to look into DNA for a RNA-seq-related task (DNA and RNA are intrinsically linked and the expression of one modality is affected by the other and vice versa). We will reemphasize these discussion points in the final version of the paper.

---------

Overall, we hope these changes address the different concerns that have been raised and reflect our willingness to improve the paper in all possible ways. Please, do not hesitate to provide further feedback.

---

### Decision · Program_Chairs · 2024-09-25

**Decision:**

Accept (poster)

**Comment:**

This work introduces IsoFormer, a multimodal approach that leverages pre-trained models on DNA, RNA, and protein sequences to predict transcript isoforms across multiple human tissues. The paper employs a modified cross-attention mechanism and concatenation scheme to create multi-modal embeddings, achieving state-of-the-art performance in isoform prediction. Throughout the review process, the benchmarking experiments and presentations have been refined and significantly improved. While the methods themselves may lack novelty, the introduction of the new challenge of predicting isoforms, combined with the availability of relevant data, will be of strong interest to NeurIPS audience.